# The Influence of Locality on Phenolic Profile and Antioxidant Capacity of Bud Extracts

**DOI:** 10.3390/foods10071608

**Published:** 2021-07-12

**Authors:** Zuzana Kovalikova, Jan Lnenicka, Rudolf Andrys

**Affiliations:** 1Department of Biology, Faculty of Science, University of Hradec Kralove, Rokitanskeho 62, 500 03 Hradec Kralove, Czech Republic; jan.lnenicka@uhk.cz; 2Department of Chemistry, Faculty of Science, University of Hradec Kralove, Rokitanskeho 62, 500 03 Hradec Kralove, Czech Republic; rudolf.andrys@uhk.cz

**Keywords:** gemmotherapy, phenolic acid, antioxidants, UHPLC-MS (ultra-high-pressure liquid chromatography)

## Abstract

Gemmotherapy represents the most recent therapeutic technique that uses the properties of extracts from fresh meristematic plant tissues, mainly buds and sprouts, by macerating them in ethanol and glycerol. The harvesting time and the location can significantly affect the chemical composition of the buds. Therefore, this work aimed to point out the possible variability in the phenolic content and the antioxidant potential of extracts prepared from commonly grown trees in the Czech Republic. Extracts from buds collected during autumn and spring in three different localities were analysed using UHPLC-MS (ultra-high-pressure liquid chromatography) for the phenols profile. Five tests assays were used for the evaluation of the extract antioxidant potential. The sampling time positively affected the content of total phenols, flavonoids, and phenolic acids. The increased levels of total phenols and flavonoids in localities with high and medium pollution may be the result of the higher levels of NO and SO_2,_ the main air pollutants. However, surprisingly, the content of phenolic acid showed the highest values in the area with the lowest pollution. The results of antioxidant tests did not completely correlate with the levels of phenolic metabolites, which may be due to the involvement of other active molecules (e.g., ascorbate, tocopherol, or proline) in the antioxidant machinery.

## 1. Introduction

Phenolic compounds represent common secondary metabolites in vascular plants. They exhibit great structural diversity and play an important role in the defence responses against various environmental stimuli, such as ultraviolet radiation, heavy metal pollutions, or plant–insect interactions. They are present in all plant organs and are therefore an integral part of the human diet. Polyphenols have drawn increasing attention due to their wide distribution in food, beverages, phytotherapeutics, and potent antioxidant properties [1].

Antioxidants are compounds that can delay, inhibit, or prevent the oxidation of oxidizable materials by scavenging free radicals and diminishing oxidative stress. The imbalance between the excess of reactive oxygen/nitrogen species and endogenous antioxidants leads to oxidative stress, and subsequently to the development of chronic degenerative diseases, such as cancer, atherosclerosis, diabetes mellitus, rheumatism, cardiovascular diseases, or inflammatory injury [1,2].

In recent years, the demand for traditional and alternative medicine products that use the power of substances naturally contained in plants has increased. One such phytotherapeutic method is gemmotherapy, which uses more biologically active substances in the embryonic parts of plants rather than the grown parts of plants. Here, the extracts are obtained from fresh buds and other meristematic tissues (young sprouts, leaves, or roots) macerated in a mixture of water, alcohol, and glycerol. The composition of such extracts is more complex because, as well as secondary metabolites, it consists of vitamins, proteins, amino acids, growth factors, hormones, or cytokines, and, therefore, acts on the body at various physiological levels, usually more effectively than common herbal tinctures [3]. Due to the good availability of macerating agents and the easiness of extract preparation, the production of gemmotherapeutic preparations is becoming accessible to the general public. The quality of these preparations may be influenced by the genotype and variety of the plant species, the phenological stage of the buds, and the environmental characteristics of the sampling locality [1,4,5,6]. Therefore, this work aimed to point out the possible variability in the phenolic content and the antioxidant potential of extracts prepared from commonly grown trees in the Czech Republic, namely birch, oak, and maple. The same analyses were performed on commercial preparations that are the best-selling in the Czech market. On account of their properties, birch and oak extracts are used as a drainage agent and are included at the beginning of phytotherapeutic treatment.

Birch (*Betula* spp. L.) leaves and other parts, i.e., buds, bark, or essential oil, are traditionally used for healing urinary disorders, skin diseases, or rheumatism. Triterpenes (betulin, betulinic acid, and lupeol), tannins, and flavonoids, mainly quercetin and hyperoside glycosides, are the main components and showed antioxidant and antimicrobial activity [1,6,7]. Oak (*Quercus* L.) bark is widely used in traditional folk medicine due to its antiphlogistic and antimicrobial properties. Leaf, bark, or acorn extracts also showed antioxidant activity [8,9]. A total of 50 species of the genus *Acer* L., commonly known as maple, have been used in traditional medicine. Phenylpropanoids, flavonoids, tannins, terpenoids, and diarylheptanoids are the most abundant and major bioactive constituents showing antioxidant, antitumor, antimicrobial, anti-inflammatory, and antidiabetic activities [10].

## 2. Materials and Methods

### 2.1. Chemicals and Reagents

All of the solvents, reagents, and standards used were of analytical grade. Ethanol 96% (*v*/*v*), glycerol, Folin–Ciocalteu’s phenol reagent, sodium carbonate anhydrous, hexahydrate aluminum chloride, hide powder, DPPH (2,2′-diphenyl-1-picrylhydrazyl), ABTS (2,2′-azino-bis(3-ethylbenzothiazoline-6-sulfonic acid)), PMS (N-methylphenazonium methyl sulfate), NADH (β-nicotinamide adenine dinucleotide reduced), NBT (nitrotetrazolium blue chloride), SNP (sodium nitroprusside), L-ascorbic acid, 2-deoxy-D-ribose, EDTA, hexahydrate iron(III) chloride, hydrogen peroxide 35% (*v*/*v*), sulfanilamide, N-(1-naphthyl)ethylenediamine dihydrochloride, TCA (trichloroacetic acid), and TBA (2-thiobarbituric acid) were purchased from Sigma-Aldrich (Steinheim, Germany). 

The chemical reagent and reference standards for UHPLC (ultra-high-pressure liquid chromatography), including formic acid, acetonitrile, protocatechuic acid, *p*-hydroxybenzoic acid, caffeic acid, chlorogenic acid, *t*-cinnamic acid, ferulic acid, gallic acid, *p*-coumaric acid, salicylic acid, syringic acid, and vanillic acid were produced by Sigma-Aldrich (Steinheim, Germany).

### 2.2. Plant Material and Sample Preparation

Buds of three different trees, *Acer pseudoplatanus* L., *Betula pendula* Roth, and *Quercus robur* L., that originated from three different localities with different levels of environmental impact were used. The plant material was collected in December 2019 and in April 2020, when the buds began to open (Figure 1). The pollution degree was estimated on the basis of the pollution maps available on the website of the Czech Meteorological Institute [11]. The measurement values of the following pollutants from surrounding sources of pollution were included in the evaluation: NO_x_, carbon monoxide, sulphur dioxide, CFCs, PM, heavy metals, and VOC. Jičín (J; GPS: 50°27′34″ N 15°26′42″ E; altitude 325 m) represented the locality with the lowest pollution impact. The vegetation around the blind branches of the Elbe in Hradec Králové (HK; GPS: 50°11′17″ N 15°49′13″ E; altitude 255 m) represented the locality of common pollution. Artificial plantings with occasional rejuvenation around the Opatovice n. Labem (O; GPS: 50°7′10″ N 15°47′30″ E; altitude 220 m) thermal power station were selected as the polluted location. For more details, see Appendix A. Commercial products from two different Czech herbal companies were analysed: the company Naděje (N), located in Brodek u Konice (altitude 600–680 m), and the company Rabštejnská Apatyka (RA), located in Srní (declared sampling in Nation Park Šumava; altitude 800–1250 m). The environmental impact on these two localities is minimal. 

The extraction method followed the protocol for bud preparation detailed in the European Pharmacopoeia [12]. The mother extract solution was prepared macerating one part fresh buds and 20 parts solution containing 42% *v*/*v* ethanol and 25.5% *v*/*v* glycerol. After four weeks of cold maceration (in a dark place at laboratory temperature), extracts were filtered (Whatman Filter Paper, London, UK) and diluted with the same solution at a ratio of 1:10 and used for the following analyses. 

### 2.3. Estimation of Phenolic Compounds

#### 2.3.1. Total Phenolic, Tannins, and Flavonoids Content

The content of total phenols was determined spectrophotometrically (Cintra 101, Dandenong, Australia) using the Folin–Ciocalteu method with gallic acid as a standard. Briefly, 30 µL of the sample extract, 470 µL of distilled water, 975 µL 2% (*w*/*v*) sodium carbonate, and 25 µL of Folin–Ciocalteu reagent solution were incubated for 60 min at 45 °C. After cooling, the absorbance was read at 750 nm [13]. 

The aluminium chloride method, with quercetin as a standard, was used for the determination of the total flavonoids level [13]. The sample extract (500 µL) was mixed with 500 µL of 2% aluminum chloride (*w*/*v*, diluted with methanol) and then incubated at room temperature for 60 min. The absorbance was read at 420 nm. 

#### 2.3.2. Determination of Phenolic Acid

The contents of phenolic acids were determined by UHPLC on Zorbax RRHD Eclipse plus C18 column (2.1 × 50 mm, 1.8 µm) (Agilent) with a 6470 Series Triple Quadrupole mass spectrometer (Agilent) (electrospray ionization in negative ion mode) as a detector. Eluents: (A) 0.05% formic acid in water and (B) 0.05% formic acid in acetonitrile were used in the following gradient program: 0–1 min (5% B), 2.0–4.0 min (20% B), 8.0–9.5 min (70% B), and 10.0–11.0 min (5% B). The MS source conditions were as follows: gas temperature 350 °C, gas flow 9 L min^−1^, nebulizer 35 psi, sheath gas temperature 380 °C, sheath gas flow 12 L min^−1^, capillary 2500 V, and nozzle voltage 0 V. Selected MRM transitions were followed for each compound: protocatechuic acid (153.0 => 109.0, 91.0), *p*-hydroxybenzoic acid (137.0 => 108.0, 92.0), caffeic acid (179.0 => 135.0, 107.0), chlorogenic acid (353.1 => 191.0, 127.0), *t*-cinnamic acid (147.1 => 103.0, 77.0), ferulic acid (193.1 => 134.1, 178.0), gallic acid (169.0 => 125.0, 119.0), *p*-coumaric acid (163.1 => 119.0, 104.9), salicylic acid (137.0 => 93.0, 65.0), syringic acid (197.1 => 182.0, 123.0), and vanillic acid (167.0 => 152.0, 108.0) [14]. A representative chromatogram is shown in Figure 2.

### 2.4. Antioxidant Activity

#### 2.4.1. DPPH Free Radical Scavenging Assay

Bud extracts were assayed by the discoloration of a solution of DPPH· as previously reported [15] with some modifications. The mixture of 60 µM methanolic solution of DPPH· (1.5 mL) and the sample extract (10 µL) was left in the dark for 30 min and evaluated spectrophotometrically at 517 nm. The percentage of DPPH· scavenging effect was calculated using the formula:% Scavenging = [(A_control_ − A_sample_)/A_control_] × 100(1)
where A_sample_ is the absorption of the solution with extract, and A_control_ is the absorbance of the solution without extract. 

#### 2.4.2. ABTS Radical Decoloration Assay

The ABTS^+^ assay was performed by bleaching the cationic radical ABTS^+^ as described by Soto et al. [16]. A solution of 7 mM ABTS^+^ and 2.5 mM potassium persulfate was left to stabilize from 12 to 16 h in the dark at room temperature before use. Afterward, the solution was diluted with methanol (80%, *v*/*v*) until an initial absorbance of 0.70 ± 0.02 was obtained at 734 nm. The sample extract (10 µL) was mixed with 300 µL of prepared working ABTS^+^ solution in a 96-well plate and incubated for 6 min in the dark at room temperature. The absorbance was measured at 734 nm in a microplate reader (SPARK^®^ Tecan Trading AG, Männedorf City, Switzerland). The scavenging effect was calculated using the formula:% Scavenging = [(A_control_ − A_sample_)/A_control_] × 100(2)
where A_sample_ is the absorption of the solution with extract, and A_control_ is the absorbance of the solution without extract.

#### 2.4.3. Hydroxyl (OH) Radical Scavenging Assay

For the generation of the hydroxyl radicals, the deoxyribose method was used. The mixture of the sample extract (10 μL), potassium phosphate buffer (780 μL; 10 mM, pH 7.4), ascorbic acid (10 μL; 0.1 mM), FeCl_3_ (50 μL; 0.04 mM), H_2_O_2_ (50 μL; 2.13 mM), and deoxyribose (100 μL; 2.8 mM) was incubated in a water bath for 60 min at 37 °C. Then, 1.0 mL of 2.8% TCA (*w*/*v*) and 1.0 mL of 1% TBA (*w*/*v*) were added and heated in a water bath for 15 min at 100 °C. After cooling the absorbance was read at 532 nm. The percentage of deoxyribose degradation was calculated using the formula:% Inhibition ·OH = [(1 − A_sample_/A_control_)] × 100(3)
where A_sample_ is the absorption of the solution with extract, and A_control_ is the absorbance of the solution without extract [15].

#### 2.4.4. Superoxide Anion Radical Scavenging Activity (O_2_^−^)

Superoxide radicals were generated by the NADH/PMS system according to the previously described procedure [17]. The sample extract (50 µL) was mixed with 50 µL NADH (166 µM), 150 µL NBT (43 µM), and 50 µL PMS (2.7 µM) in a 96-well plate and incubated for 2 min at room temperature. All components were dissolved in a phosphate buffer (19 mM, pH 7.4). The absorbance was measured at 560 nm. The percentage of scavenging effect was calculated using the formula:% Scavenging O_2_^−^ = [(1 − A_sample_)/A_control_] × 100(4)
where A_sample_ is the absorption of the solution with extract, and A_control_ is the absorbance of the solution without extract.

#### 2.4.5. Nitric Oxide Scavenging Assay (NO)

The activity was determined spectrophotometrically in a 96-well plate reader. The reaction mixtures consisted of the sample extract (100 µL) and 100 µL SNP (20 mM) were preincubated for 60 min at 25 °C under light exposure. Then, 100 µL of Griess reagent was added, and the absorbance was read at 540 nm. The percentage of scavenging effect was calculated using the formula: % Scavenging NO = (A_sample_/Ac_ontrol_) × 100(5)
where A_sample_ is the absorption of the solution with extract, and A_control_ is the absorbance of the solution without extract.

### 2.5. Data Processing

Statistica 10.0 (StatSoft Inc., Tulsa, OK, USA) software was used for statistical analyses. The comparison of differences in the experiment was based on the one-way analysis of variance (ANOVA) and Tukey’s test at the significance level *p* < 0.05. Six individual samples were used for analyses of each parameter. 

## 3. Results

The overall phytochemical profile of the analysed material, phenolic compounds included, is influenced by several factors, such as the genetic origin of the plant, or time and technique of sample collection, and the processing of the material. In addition, environmental conditions (influence of biotic and abiotic factors) significantly affect the qualitative and quantitative composition of plant materials and thus the possible therapeutic potential of medicinal products [1,6]. Therefore, in our study, we analysed the plant material harvested in two time periods (autumn and spring), originating from three localities that differed in the degree of pollution, and compared our results with commercially available preparations. 

Based on the results shown in Table 1, it is clear that the time of sampling affected the content of total phenols, flavonoids, and phenolic acids (given as a sum). In all trees, the values were significantly higher in the extracts from the buds collected in the spring. The only exception was in oak, where the sum of phenolic acid decreased depending on the time of harvest. Therefore, only spring extracts were used for further analyses.

The changes in metabolite content, with respect to sampling time, may be related to the transition of the buds from the dormant phase to the actively growing phase, where the need for metabolites is different. In birch, the content of hydrolysable tannins (gallotannin and ellagitannins) and flavonoid aglicones decreased by up to 90%, depending on the bud transformation to the adult leaf. The content of phenolic acids (mainly hydroxycinnamic acid derivatives) increased [6]. A similar increase in hydroxycinnamic acids and gallic acid, with respect to the time of harvest, was also observed in four tested *Castanea* species. The parallel increase in flavonoids, namely quercetin and rutin, and tannins also increased [4]. The work of Varigi et al. [5] pointed out that the effect of the season was considerably greater than that of the genotype, ontogenetic stage, and location. Here, the content of glycosylated flavonoids, epigallocatechin and epicatechin, decreased with the ontogenesis of the blackcurrant bud. Concentration variability, with respect to sampling time, was also observed for other secondary metabolites, for example, the content of terpenes in the essential oil of six blackcurrant cultivars decreased with the disruption of bud dormancy [18].

The monitored trees responded differently to the environmental impact (Figure 3). For maple and birch, the highest values of total phenols (32.14 and 48.46 mg g^−1^ FW, respectively) and flavonoids (7.14 and 16.73 mg g^−1^ FW, respectively) were observed in bud extracts in the slightly contaminated locality HK. In the case of oak, the highest values were in the clean locality J. The commercially available extracts of the monitored trees alone showed significantly lower values compared to our measurements. In the case of phenols, there were no significant differences between N and RA producers, 9.39 and 9.69 mg g^−1^ FW for maple, 16.83 and 19.21 mg g^−1^ FW for birch, and 11.75 and 13.48 mg g^−1^ FW for oak, respectively. The content of flavonoids in the maple extract of the company N (6.72 mg g^−1^ FW) showed a similar level as the extracts from the localities J and HK. On the contrary, the values in the RA extract (4.92 mg g^−1^ FW) were comparable with the locality O, and at the same time significantly lower from the above. For birch, the values in N and RA were significantly lower compared to our extracts, 6.36 and 5.63 mg g^−1^ FW, respectively. For oak, the commercial extracts showed similar values as HK, namely 3.10 mg g^−1^ FW for N and 3.08 mg g^−1^ FW for RA.

Phenolic acids can be divided into two classes: derivatives of hydroxybenzoic acid such as gallic acid, and derivatives of hydroxycinnamic acid (HCA), such as caffeic, chlorogenic, or ferulic acid [1]. Here, we followed the changes of five HCA and six hydroxybenzoic acids. Overall, the highest content of monitored phenolic acids was in birch extracts, with the exception of gallic acid. Here, the highest values were in maple extracts. In maple and oak extracts (Table 2 and Table 3), the highest values of all of the monitored HCA were recorded in the clean locality J. In the case of the HK and O sites, the values were similar. In birch (Table 4) the content of HCA was highest in J (with the exception of chlorogenic acid—locality HK)

For hydroxybenzoic acid, the effect of the sampling locality was more variable, in contrast to HCA. For maple (Table 2) the highest values were in localities J (*p*-hydroxybenzoic and salicylic acids) and HK (gallic, syringic, and vanillic acids). For oak (Table 3), the maximal amounts were in J, with the exception of syringic and vanillic acids, which had a maximum in the HK samples. Monitored values for birch varied considerably and it is not possible to draw a clear conclusion about the influence of the locality.

A study focusing on changes in the phenolic metabolism in birch leaves showed that the contents of phenolics declined with an increase in distance from the smelter. However, the changes in individual groups had a different pattern of temporal variation. Gallic acid derivatives increased with an increase in pollution, HCA derivatives showed no variation among the study site, and for flavonoids, strong variation was observed [19]. Pasqualini et al. [20] pointed out that simple phenols could also be used as biological indicators of air pollution. A negative correlation between the content of total phenolics correlated negatively with NO_x_ and positively with SO_2_ concentrations. Moreover, *p*-coumaric acid, syringic acid, and *p*-hydroxybenzoic acid concentrations increase with exposure to NO pollution, whereas gallic acid decreases in the presence of SO_2_. The highest values of the mentioned air pollutants in our study were measured at the locality O. However, similar accumulation trends were not observed. 

The causes of changes in the accumulation of individual metabolites can be seen in their biosynthetic pathway. Cinnamic acid, the common precursor of phenolic compounds, is synthesised via the shikimic acid pathway. Various chemical reactions e.g., aromatic hydroxylation, β-oxidation, and methoxylation lead to the creation of a set of HCA derivatives along with chlorogenic acids. Gallic acid, a precursor of hydrolysable tannins, is formed through an intermediate from the shikimate pathway [1]. The level of precursors and the sensitivity of enzymes in individual steps in biosynthetic pathways may be one of the factors of variation in plant environmental responses. 

Another factor influencing the overall composition and contents of metabolites may be the altitude at which the plants grow. Several meteorological factors, such as temperature, precipitation, or light intensity are closely related to the altitudinal gradient of a specific locality. The content of phenolics in elderberry leaves was significantly affected by the altitude. The increase of HCA was not gradual along the altitudinal gradient (200–1050 m) but, nonetheless, their levels were higher at the hilltop than at the foothill. The authors suggest that higher HCA values are connected with higher light intensity (especially UV-B radiation) [21]. In *Buxus* leaves, there was no clear correlation between altitude (400–1700 m) and the content of total phenols and flavonoids. However, phenolic acids were positively influenced strongly at the highest sampling locality. Thus, the authors suggest that a greater influence on the seasonal and altitudinal variation, as opposed to ambient UV-B, would have other developmental or abiotic (e.g., temperature, precipitation, or wind speed) factors [22]. On the contrary, high precipitations at low altitudes (80–200 m), in contrast to dry higher altitudes (450–700 m), for growing sites resulted in a higher synthesis of phenolic compounds [23]. The localities studied by us are in a relatively narrow range of altitude (220–330 m), so we believe that this does not play a significant role.

From the above, it is clear that the overall qualitative and quantitative composition of phenols can be affected by several factors simultaneously. Studies evaluating several variables simultaneously already differ in determining the main factor influencing the level of phenolic substances. Moreover, constitutive genetic differences may affect the physiology of the studied species more than the different environmental conditions [5,24,25].

The values of individual phenolic metabolites in commercially available preparations (N and RA) vary considerably in comparison with our extracts. From the above, as the possible causes, we can state that they are the sampling locality, its altitude, or the genotype of the plant. Producers declare that the plant material comes from clean sites without environmental pollution or directly from their own planting (genotype selection may take place here). Similar to our results, commercial preparations of the same species of *Rubus* and *Ribes* from different companies showed a similar percentage of individual bioactive compounds, but the values differed from laboratory-prepared extracts [25].

Studies showing the relationship between antioxidant-rich food intake and the occurrence of some chronic diseases have increased the interest in studying natural antioxidants. Some authors recommend that several tests be used in the testing of antioxidant potential that involve various mechanisms of chemical reaction, such as hydrogen atom transfer (HAT) or single electron transfer (SAT) [26]. Therefore, the extracts prepared from tree buds were subjected to five tests differing in scavenging mechanism to evaluate their overall antioxidant potential, namely the DPPH· and ABTS· test with the HAT pattern and ·OH, O_2_^−^·, and NO· tests with the SAT pattern. The results are expressed as the % of radical scavenging. The results are shown in Table 5.

The scavenging of the DPPH radical was very effective for all trees, more than 90%. Significant differences between localities were observed only for maple from the HK locality, where the overall activity was the lowest, 95.15%. Strong antioxidant activity, more than 90%, was also observed for the ABTS test. The results of the ·OH scavenging test showed lower values, ranging from 85–90% in comparison with previously mentioned tests. Significant differences between localities were found only in maple extracts. Overall, the highest O_2_^−^ scavenging activity was recorded in birch extracts, specifically in the HK locality. Locations J and O did not differ significantly. In oak and maple, the highest levels were observed in the HK extracts (87% and 90%, respectively). Based on the NO· assay, birch and oak extracts showed the highest activity in locality J; 68% and 79%, respectively. For maple extracts, it was the locality O. 

Similar to our results, aqueous and methanol extracts of birch leaves showed high antioxidant activity, more than 80%, measured via the DPPH and ABTS test [27]. However, when compared to the activity of certain phenolic acids, significantly higher concentrations of birch extracts were needed to achieve similar antioxidant activity [28]. The high efficacy of phytotherapeutic extracts was demonstrated in the work of Raiciu et al. [29], where the antioxidant capacity of over 90% birch, *Salix*, and *Ribes* extracts was monitored even after considerable dilution of the extract (up to 100-fold). 

The different antioxidant activities between tree extracts may be due to the variability of composition, content, and chemical character of various active compounds, and the synergy between them and other natural substances [1]. The study comparing the composition of water bark extracts of alder, pine, and oak reveals the higher contents of phenolic biologically active components (phenols, flavonoids, tannins) and radical scavenging activity were in samples collected in the city with medium pollution from continental climatic zones compared to samples from the clean environment of a coastal natural park [30]. Variability, with respect to genetic structure and environmental condition, was also shown in the *Alcea* species in the content of total phenolics, flavonoids, anthocyanins, and overall antioxidant activity [31]. A significant effect of climatic zones on the phenolic content and antioxidant potential of the *Aloe vera* plant was monitored. Extracts of plants from the highland and semi-arid zones of northern India possessed maximum antioxidant potential compared to the tropical zones of southern India [32]. 

Generally, the radical scavenging activity of phenolic acids depends on the number and position of hydroxyl (−OH) groups and methoxy (−OCH_3_) substituents in the molecules. Caffeic acid, most often esterified with quinic acid as in chlorogenic acid (both have two −OH groups), gallic acid (three −OH groups), and ferulic acid (one −OH and one −OCH_3_), showed substantial antioxidant properties to scavenge free radicals [15,33,34]. Thus, higher concentrations of these phenolic acids can be reflected in the antioxidant activity of the extracts. In our samples, the highest concentrations of chlorogenic, caffeic, and ferulic acids were recorded in birch extracts, which, however, was manifested only in the cases of the O_2_^−^ and NO· tests. However, when we focus on the correlation between the content of the mentioned acids and the antioxidant activity within one tree, it is not possible to draw a specific conclusion about the possible influence of the studied locality. 

The preparation of gemmotherapeutic extracts is an undemanding method and thus easily accessible to the general public. Weaker botanical knowledge can lead to the easy confusion of individual species, which can also be reflected in the total content of phenolic substances and thus the possible antioxidant potential of the extracts. For example, Meda et al. [35] showed that the extracts prepared from red maple contained higher amounts of total phenols, flavonoids, and anthocyanins than sugar maple, which was also reflected in the higher antioxidant activity measured by the DPPH test. Similar results were recorded in the study of two oak species, where leaf extracts of *Q. saliciana* showed the maximum inhibition activities in ABTS radical scavenging assays, however, in the DPPH test *Q. serrata* leaf extracts showed better results [36].

## 4. Conclusions

Based on the data, we can conclude that the time of bud sampling positively affected the content of total phenols, flavonoids, and phenolic acids. The influence of environmental pollution, as another studied variable, was variable among the studied trees. Increased levels of total phenols and flavonoids in localities O and HK may be the result of higher levels of NO and SO_2_, the main air pollutants. However, surprisingly, the content of phenolic acid showed the highest values in the area with the lowest pollution. The influence of other variables, such as altitude, climatic conditions (e.g., temperature, precipitation, wind), and the genetic origin of plants can play an important role. From the above, there is a need for larger and longer studies, that take into account several of the above variables.

## Figures and Tables

**Figure 1 foods-10-01608-f001:**
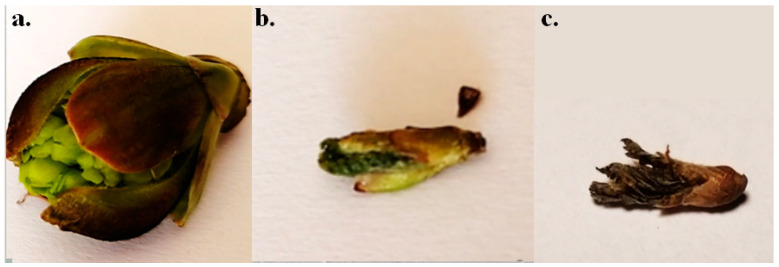
Buds of individual trees collected in April 2020; (**a**). *Acer pseudoplatanus*, (**b**). *Betula pendula*, (**c**). *Quercus robur*.

**Figure 2 foods-10-01608-f002:**
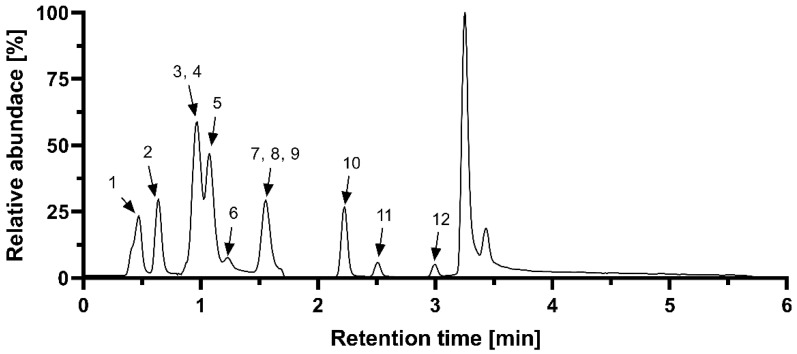
Representative chromatogram of measured phenolic acids. 1. gallic acid; 2. protocatechuic acid; 3. chlorogenic acid; 4. salicylic acid; 5. *p*-hydroxycbenzoic acid; 6. vanillic acid; 7. caffeic acid; 8. syringic acid; 9. benzoic acid; 10. *p*-coumaric acid; 11. ferulic acid; 12. cinnamic acid.

**Figure 3 foods-10-01608-f003:**
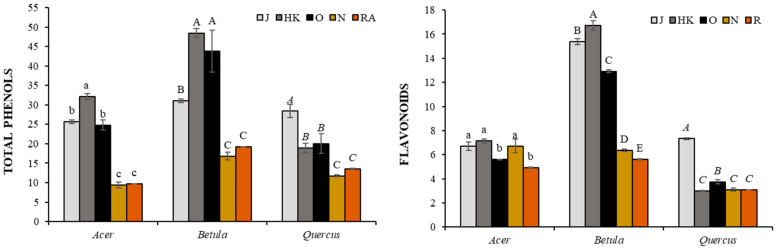
The content of total phenols (mg g^−1^ FW) and flavonoids (mg g^−1^ FW) in spring bud extracts of *Acer pseudoplatanus*, *Betula pendula*, and *Quercur robur*. J, Jičín; HK, Hradec Králové; O, Opatovice n. Labem; N, Naděje; RA, Rabštejnská Apatyka. Data are as mean ± SDs (*n* = 6). Values within column followed by the same letter(s), are not significantly different according to Tukey’s test (*p* < 0.05). The same type of letters shows a significant difference between the values of individual tree.

**Table 1 foods-10-01608-t001:** The content of total phenols (mg g^−1^ FW), flavonoids (mg g^−1^ FW), and sum of phenolic acids (µg g^−1^ FW) in bud extracts of *Acer pseudoplatanus, Betula pendula,* and *Quercur robur*. J, Jičín; HK, Hradec Králové; O, Opatovice n. Labem. Data are as mean ± SDs (*n* = 6). Values within column followed by the same letter(s), are not significantly different according to Tukey’s test (*p* < 0.05).

		Soluble Phenols (mg g^−1^ FW)	Flavonoids (mg g^−1^ FW)	Phenolic Acid-Sum (µg g^−1^ FW)
*Acer pseudoplatanus*				
J	Autumn Spring	17.66 ± 2.27 c 25.67 ± 0.41 b	2.99 ± 0.05 d 6.70 ± 0.34 a	202.10 ± 3.47 d 425.71 ± 11.85 b
HK	Autumn Spring	20.44 ± 3.34 bc 32.14 ± 0.80 a	4.01 ± 0.10 c 7.14 ± 0.16 a	122.00 ± 7.33 e 960.96 ± 29.77 a
O	Autumn Spring	22.25 ± 24.81 bc 24.81 ± 1.26 bc	2.42 ± 0.01 e 5.60 ± 0.01 d	169.13 ± 2.91 d 355.87 ± 13.89 c
*Betula pendula*				
J	Autumn Spring	26.72 ± 1.02 c 31.04 ± 0.43 bc	12.38 ± 0.96 c 15.37 ± 0.24 ab	412.03 ± 4.61 c 591.82 ± 11.37 b
HK	Autumn Spring	30.06 ± 0.84 bc 48.46 ± 1.19 a	14.84 ± 0.21 b 16.73 ± 0.38 a	599.99 ± 7.46 b 720.67 ± 17.09 a
O	Autumn Spring	33.88 ± 1.18 b 43.85 ± 5.37 a	10.42 ± 0.96 c 12.91 ± 013 c	355.57 ± 10.09 d 435.43 ± 6.78 c
*Quercus robur*				
J	Autumn Spring	21.94 ± 0.23 b 28.40 ± 1.68 a	2.63 ± 0.28 c 7.33 ± 0.06 a	288.85 ± 4.28 a 234.63 ± 4.11 b
HK	Autumn Spring	14.47 ± 0.54 c 18.89 ± 1.24 b	1.95 ± 0.19 d 2.99 ± 0.04 c	109.9 ± 3.85 c 92.23 ± 0.64 d
O	Autumn Spring	14.06 ± 0.43 c 20.01 ± 2.56 b	1.60 ± 0.13 d 3.75 ± 0.18 b	103.35 ± 4.10 c 83.91 ± 1.73 d

**Table 2 foods-10-01608-t002:** The content of phenolic acids (µg g^−1^ FW) in bud extracts of *Acer pseudoplatanus*. Values are expressed as mean ± SDs (*n* = 5). Values within lines, followed by the same letter(s), are not significantly different according to Tukey’s test (*p* < 0.05).

*Acer pseudoplatanus*					
	Jičín	Hradec Králové	Opatovice n. Labem	Naděje	Rabštejnská Apatyka
hydroxycinnamic acids					
*t*-caffeic acid	11.943 ± 0.589 a	2.438 ± 0.113 c	3.606 ± 0.092 b	3.904 ± 0.201 b	4.029 ± 0.026 b
chlorogenic acid	14.863 ± 1.075 a	2.296 ± 0.189 d	5.971 ± 0.074 c	8.504 ± 0.263 b	7.171 ± 0.443 bc
cinnamic acid	6.618 ± 0.577 a	2.635 ± 0.016 c	2.497 ± 0.146 c	5.429 ± 0.165 b	5.053 ± 0.139 b
*p*-coumaric acid	13.962 ± 0.461 b	5.323 ± 0.436 c	5.564 ± 0.147 c	17.326 ± 0.501 a	17.794 ± 0.739 a
ferulic acid	6.560 ± 0.256 a	5.007 ± 0.209 c	1.362 ± 0.129 d	4.957 ± 0.163 c	5.676 ± 0.356 b
hydroxybenzoic acids					
protocatechuic acid	0.785 ± 0.053 bc	0.697 ± 0.044 c	0.812 ± 0.025 ab	0.889 ± 0.022 ab	0.799 ± 0.013 ab
*p*-hydroxybenzoic acid	0.893 ± 0.171 a	0.691 ± 0.066 a	0.668 ± 0.138 a	0.722 ± 0.071 a	0.732 ± 0.005 a
gallic acid	364.29 ± 12.88 b	937.63 ± 29.63 a	332.57 ± 13.63 b	156.74 ± 5.30 c	181.49 ± 3.34 c
salicylic acid	3.116 ± 0.084 b	0.392 ± 0.028 d	0.777 ± 0.035 c	4.078 ± 0.072 a	4.401 ± 0.284 a
syringic acid	0.553 ± 0.117 d	1.419 ± 0.091 a	0.636 ± 0.053 cd	0.779 ± 0.035 bc	0.841 ± 0.021 b
vanillic acid	2.130 ± 0.154 ab	2.429 ± 0.108 a	1.406 ± 0.098 b	1.453 ± 0.301 b	1.664 ± 0.492 b

**Table 3 foods-10-01608-t003:** The content of phenolic acids (µg g^−1^ FW) in bud extracts of *Quercus robur*. Values are expressed as mean ± SDs (*n* = 5). Different letters in lines show significant differences (*p* < 0.05) according to Tukey’s test.

*Quercus robur*					
	Jičín	Hradec Králové	Opatovice n. Labem	Naděje	Rabštejnská Apatyka
hydroxycinnamic acids					
*t*-caffeic acid	5.477 ± 1.303 a	2.604 ± 0.163 b	1.737 ± 0.146 b	2.255 ± 0.083 b	1.997 ± 0.150 b
chlorogenic acid	13.687 ± 0.576 a	0.551 ± 0.074 d	2.613 ± 0.150 c	2.161 ± 0.012 c	3.272 ± 0.23 b
cinnamic acid	58.520 ± 3.290 a	4.304 ± 0.765 b	7.014 ± 0.563 b	6.742 ± 0.135 b	6.659 ± 0.338 b
*p*-coumaric acid	14.117 ± 0.205 a	7.324 ± 0.305 c	11.003 ± 0.129 b	2.645 ± 0.073 d	3.223 ± 0.246 e
ferulic acid	5.913 ± 0.117 a	5.780 ± 0.744 d	9.247 ± 0.083 a	6.971 ± 0.307 bc	7.468 ± 0.385 b
hydroxybenzoic acids					
protocatechuic acid	3.567 ± 0.147 a	2.862 ± 0.288 b	1.127 ± 0.094 d	1.608 ± 0.147 c	1.642 ± 0.042 c
*p*-hydroxybenzoic acid	74.467 ± 2.800 a	17.028 ± 0.544 b	8.583 ± 0.241 d	12.951 ± 0.729 c	15.562 ± 0.554 bc
gallic acid	27.317 ± 0.686 a	33.223 ± 0.388 a	25.305 ± 0.624 a	28.018 ± 0.759 a	17.217 ± 1.115 b
salicylic acid	9.565 ± 0.385 a	3.482 ± 0.188 c	3.061 ± 0.054 c	3.461 ± 0.256 c	4.463 ± 0.375 b
syringic acid	4.788 ± 0.233 b	6.358 ± 0.358 a	6.280 ± 0.261 ac	1.525 ± 0.045 c	1.884 ± 0.112 c
vanillic acid	17.217 ± 1.115 a	8.718 ± 0.248 b	7.941 ± 0.280 bc	6.478 ± 0.464 c	6.501 ± 0.312 c

**Table 4 foods-10-01608-t004:** The content of phenolic acids (µg g^−1^ FW) in bud extracts of *Betula pendula*. Values are expressed as mean ± SDs (*n* = 5). Different letters in lines show significant differences (*p* < 0.05) according to Tukey’s test.

*Betula pendula*					
	Jičín	Hradec Králové	Opatovice n. Labem	Naděje	Rabštejnská Apatyka
hydroxycinnamic acids					
*t*-caffeic acid	38.757 ± 0.838 a	21.769 ± 1.252 c	27.718 ± 0.972 b	20.799 ± 0.156 c	26.975 ± 2.158 b
chlorogenic acid	201.22 ± 15.03 d	410.96 ± 8.62 b	128.30 ± 2.64 e	456.41 ± 3.02 a	353.76 ± 1.80 c
cinnamic acid	178.128 ± 4.251 a	117.457 ± 6.243 b	89.765 ± 2.134 c	70.220 ± 0.391 d	78.272 ± 1.954 d
*p*-coumaric acid	39.871 ± 1.735 a	22.722 ± 1.283 b	22.029 ± 1.187 b	17.334 ± 1.043 c	17.017 ± 0.648 c
ferulic acid	75.945 ± 2.667 a	74.336 ± 4.941 a	81.685 ± 1.084 a	21.606 ± 2.077 b	26.939 ± 0.848 b
hydroxybenzoic acids					
protocatechuic acid	2.148 ± 0.245 c	2.266 ± 0.155 c	3.913 ± 0.160 bc	5.596 ± 1.227 ab	7.334 ± 1.043 a
*p*-hydroxybenzoic acid	1.966 ± 0.211 c	3.039 ± 0.152 b	4.121 ± 0.531 a	2.087 ± 0.233 c	2.769 ± 0.285 bc
gallic acid	35.097 ± 0.516 c	43.095 ± 1.379 b	54.992 ± 2.760 a	12.948 ± 0.271 e	21.178 ± 1.352 d
salicylic acid	7.326 ± 0.575 c	10.122 ± 0.235 b	7.966 ± 0.317 c	16.354 ± 0.546 a	15.007 ± 0.899 a
syringic acid	1.252 ± 0.048 a	1.333 ± 0.046 a	0.918 ± 0.094 b	0.561 ± 0.030 c	0.540 ± 0.024 c
vanillic acid	10.113 ± 0.513 c	13.578 ± 0.462 b	15.647 ± 0.447 a	9.678 ± 0.394 c	9.296 ± 0.594 c

**Table 5 foods-10-01608-t005:** Antioxidant activities of bud extracts. Values are expressed as mean ± SDs (*n* = 5). J, Jičín; HK, Hradec Králové; O, Opatovice n. Labem; N, Naděje; RA, Rabštejnská Apatyka. Different letters show significant differences (*p* < 0.05) between groups for the same experiment.

	DPPH (%)	ABTS (%)	OH (%)	O_2_^−^ (%)	NO·(%)
*Acer*					
J	98.13 ± 0.25 a	95.84 ± 0.96 a	93.04 ± 0.20 a	72.28 ± 2.34 d	63.85 ± 0.36 bc
HK	95.15 ± 0.72 b	96.66 ± 0.16 a	86.95 ± 0.80 b	87.17 ± 1.10 b	79.91 ± 1.39 a
O	98.44 ± 0.20 a	97.02 ± 0.05 a	88.40 ± 0.11 b	79.73 ± 0.60 c	80.86 ± 1.93 a
N	98.16 ± 0.13 a	96.51 ± 0.38 a	84.69 ± 0.72 c	97.94 ± 1.91 a	63.36 ± 2.45 c
RA	98.77 ± 0.03 a	97.04 ± 0.13 a	88.40 ± 0.57 b	90.38 ± 0.91 b	67.75 ± 0.81 b
*Betula*					
J	98.76 ± 0.06 b	94.67 ± 2.61 ab	84.30 ± 0.69 b	93.70 ± 1.05 a	68.33 ± 0.63 b
HK	98.63 ± 0.07 b	95.94 ± 1.41 ab	86.68 ± 2.48 ab	93.01 ± 1.70 a	67.64 ± 0.58 b
O	98.31 ± 0.43 b	97.05 ± 0.05 a	89.80 ± 2.53 a	94.50 ± 1.82 a	65.42 ± 0.46 b
N	99.73 ± 0.01 a	92.28 ± 0.46 b	87.08 ± 1.30 ab	91.52 ± 1.11 a	72.42 ± 0.64 a
RA	99.58 ± 0.01 a	95.17 ± 1.16 ab	90.13 ± 0.50 a	93.13 ± 2.73 a	73.14 ± 2.65 a
*Quercus*					
J	98.60 ± 0.16 c	93.75 ± 1.64 a	86.68 ± 0.72 ab	85.22 ± 1.91 b	79.03 ± 1.81 a
HK	98.82 ± 0.13 bc	96.92 ± 0.16 a	89.73 ± 0.30 a	90.82 ± 1.70 a	66.18 ± 0.12 c
O	98.91 ± 0.08 ab	90.44 ± 0.11 a	88.34 ± 4.38 a	75.95 ± 0.60 c	63.52 ± 1.13 c
N	99.05 ± 0.08 ab	94.39 ± 2.65 a	82.44 ± 0.30 b	83.39 ± 1.89 b	59.09 ± 1.01 d
RA	99.17 ± 0.02 a	95.92 ± 1.18 a	88.14 ± 0.75 a	90.72 ± 0.62 b	70.23 ± 0.98 b

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
