# Peer review of "The Influence of Locality on Phenolic Profile and Antioxidant Capacity of Bud Extracts"

_foods, 2021, doi:10.3390/foods10071608_

Round 1
Reviewer 1 Report
I find the paper interesting and clearly presented. I suggest incorporation and examplary chromatogram to MS showing phenolic profile(s) of investigated extracts.
Author Response
Reply to Reviewer 1.
We would like to thank the Reviewer for valuable comments that helped us to improve the manuscript.
- I suggest incorporation and exemplary chromatogram to MS showing phenolic profile(s) of investigated extracts.
A: The exemplary chromatogram was added into the manuscript as a Figure 2.

Reviewer 2 Report
The manuscript is expected to be of interest in the scientific community.
However, in my opinion the authors should make some revisions, in order to improve the quality of their manuscript, as follows.
- title: as the locality (clean or polluted) seems to have no influence on the phenolic profile, I suggest to remove the word "locality" from the title
- please provide the plant full name, including the botanist who discovered or named the species after the plant scientific name (e.g. Quercus robur L.)
- line 117: it seems to me that after maceration the extracts were directly diluted 1:10 and used for all the analyses. I don't understand the rational behind
- Which extracts doses were used in the antioxidant activity? The calculation of IC50 would be useful for further comparison. Instead, if the authors directly tested the diluted extracts (at an unknown concentration, but I image it is quite high) it is very difficult to see differences in antiradical assays and it has no meaning (please, refer to table 5).
- It could be useful to add the web site of Czech Meteorological Institute as reference and a table in which all the mentioned pollution variables are listed
- I suggest to try to rationalize the results globally using statistic tool (e.g. principal component analysis). Maybe some interesting correlations would allow the authors to discuss and draw some deeper conclusions about the possible influence of the studied locality.
- why did the authors select those targeted compounds in the LC-MS analyses? Did they perform a preliminary LC-MS untargeted profile? line 159: MRM transition for kaempferol is not correct.
Some minor typos at lines 15 (UHPLH-MS-->UHPLC-MS), 58 (lueol-->lupeol), 305 (keampferol-->kaempferol)
Author Response
We would like to thank the Reviewer for valuable comments that helped us to improve the manuscript.
In following we list the changes made in the paper according to recommended revisions.
- title: as the locality (clean or polluted) seems to have no influence on the phenolic profile, I suggest to remove the word "locality" from the title
A: After harmonizing all the reviewers' opinions, on the recommendation we focused more on the influence of the locality on changes in phenolic metabolism. The following change of title is also related to this: "The influence of locality on phenolic profile and antioxidant capacity of bud extracts”.
- please provide the plant full name, including the botanist who discovered or named the species after the plant scientific name (e.g. Quercus robur L.)
A: All required full botanical names, including the botanist were including.
- line 117: it seems to me that after maceration the extracts were directly diluted 1:10 and used for all the analyses. I don't understand the rational behind
A: During the preparation of extracts, we followed the European Pharmacopoeia (for better clarity newly cited in manuscript). We also took into account information on the extracts preparation from commercially available preparations. For this reason, we diluted the prepared mother extract in ratio 1:10.
- Which extracts doses were used in the antioxidant activity? The calculation of IC50 would be useful for further comparison. Instead, if the authors directly tested the diluted extracts (at an unknown concentration, but I image it is quite high) it is very difficult to see differences in antiradical assays and it has no meaning (please, refer to table 5).
A: 1:10 The dilute extracts in ration 1:10 were used in antioxidant assays to make our observations comparable to commercially available preparations. Given that the concentration of the extract used in all tests is given in the methodology section, we do not consider it necessary to specify this in Table 5.
- It could be useful to add the web site of Czech Meteorological Institute as reference and a table in which all the mentioned pollution variables are listed
A: All necessary information regarding the pollution degree of the used localities together with clear maps of the sampling location and the main pollutants are given in the supplementary materials, Table S1.
- I suggest to try to rationalize the results globally using statistic tool (e.g. principal component analysis). Maybe some interesting correlations would allow the authors to discuss and draw some deeper conclusions about the possible influence of the studied locality.
A: No interesting correlation appear after using PCA.
- why did the authors select those targeted compounds in the LC-MS analyses? Did they perform a preliminary LC-MS untargeted profile? line 159: MRM transition for kaempferol is not correct.
A: In the analysis of phenolic acids, we focused on the most common acids contained in many plant species and also on those acids that have significant antioxidant properties.
- Some minor typos at lines 15 (UHPLH-MS-->UHPLC-MS), 58 (lueol-->lupeol), 305 (keampferol-->kaempferol)
A: Mentioned minor typos were corrected.

Reviewer 3 Report
The authors presented a research article on the influence of sampling time and locality on the phenolic profile and antioxidant capacity of bud extracts.
The topic is interesting and might be suitable for this Journal.
Yet, the conducted study is only superficial in several parts and the results presented do not add anything new to what already known about the argument.
In fact, I have some major concerns about the entire research methodology conducted by the authors. I would have focused on other elements and parameters.
The things to check and redo for a possible resubmission of this paper to another Journal in the future are listed below one by one:
ABSTRACT:
- “Gemmotherapy represents a new type of phytotherapy using extracts made from embryonic parts of plants, i.e. buds”. Yes, but not only buds. This sentence is incomplete.
INTRODUCTION:
- “Birch (Betula sp.) leaves… and antidiabetic activities” I remind the authors that the exact writing is spp for species. In addition, please write the complete botanical names of the genera cited here. Then, please write something about the pharmacological properties and phytochemical patterns of the embryonic organs of these species since the topic of your research is more focused on this. Lastly, it would not be a bad idea if you could present some data about the current existence of commercial products with the embryonic organs of species belonging to these genera. I do not want the names of the products but the quantities.
MATERIALS AND METHODS:
- Please write DPPH and ABTS as radicals as they are as well as the other necessary compounds.
Section 2.1.:
- Please write the complete botanical names of the studied species.
- Please revise this part as write it in a discursive manner and not as hints.
- “Buds of three different trees: Acer pseudoplatanus, Betula pendula, and Quercus robur, originated from three different localities with different levels of environmental impact…” What do you mean with different levels of environmental impact? You must specify it here.
Section 2.3.:
- “The contents of phenolic acids and selected flavonoids were determined by UHPLC on Zorbax RRHD Eclipse…” Selected flavonoids? What does this mean? Does it mean that you focused on only some flavonoids? If so, this is not fine. You may have not identified a myriad of other compounds of this kind.
RESULTS:
General concern:
- All you presented here is not actually a novelty. It is obvious that the best results are given by embryonic organs in Spring since this is the right time when to collect them. You did not add anything new. In fact, my question is: why make this autumn collection? I think your work should have been focused mainly on the environmental and locality influence than on the collection time of the plant material. You would have done better to make a study more related to the interaction between buds and the environment.
- And what about the altitudes of the collection sites? Was it important?
- And what about the correlation between altitude and the biosynthesis of secondary metabolites? I think you are aware that also altitude affects this biosynthesis. Was it important, too?
- After this, how is pollution correlated with the biosynthesis of your studied secondary metabolites? This must be studied and discussed.
- Also, the decrease of some secondary metabolites from autumn to spring can be simply explained. By the way, you have embryonic organs which in Spring did not have enough time to fully biosynthesize flavonoids, tannins, and other classes. In this sense, why did you study these exact classes of natural compounds? Actually, I would not choose to collect embryonic organs if I were interested in isolating and identifying flavonoids, tannins and so on.
- How do you explain all the antioxidant results you reported? What is the relation between the collection site and the antioxidant activities? In addition, according to the phytochemical pattern you reported, I would have expected opposite results for what concerns the antioxidant activities. How do you explain them?
CONCLUSIONS:
- This section should be revised. You did not fully demonstrate your conclusions.
Author Response
Reply to Reviewer 3.
We would like to thank the Reviewer for valuable comments that helped us to improve the manuscript.
In following we list the changes made in the paper according to recommended revisions.
- ABSTRACT:
- “Gemmotherapy represents a new type of phytotherapy using extracts made from embryonic parts of plants, i.e. buds”. Yes, but not only buds. This sentence is incomplete.
A: The sentence was corrected: “Gemmotherapy represents a new type of phytotherapy using extracts made from embryonic parts of plants, i.e. buds, young sprout, leaves, or roots.”
- INTRODUCTION:
- “Birch (Betula sp.) leaves… and antidiabetic activities” I remind the authors that the exact writing is spp for species. In addition, please write the complete botanical names of the genera cited here. Then, please write something about the pharmacological properties and phytochemical patterns of the embryonic organs of these species since the topic of your research is more focused on this. Lastly, it would not be a bad idea if you could present some data about the current existence of commercial products with the embryonic organs of species belonging to these genera. I do not want the names of the products but the quantities.
A: The botanical names of the individual species were modified as recommended.
At that time, there are four companies in the Czech Republic producing gemmotherapeutic preparations from their own resources (even with a development abroad, including the USA and Russia). We added information about the analysis of the products from two best-selling companies to the manuscript and compared them with our data.
- MATERIALS AND METHODS:
- Please write DPPH and ABTS as radicals as they are as well as the other necessary compounds.
Section 2.1.:
- Please write the complete botanical names of the studied species.
- Please revise this part as write it in a discursive manner and not as hints.
A: All recommended changes have been included in the manuscript.
- “Buds of three different trees: Acer pseudoplatanus, Betula pendula, and Quercus robur, originated from three different localities with different levels of environmental impact…” What do you mean with different levels of environmental impact? You must specify it here.
A: All necessary information regarding the pollution degree of the used localities together with clear maps of the sampling location and the main pollutants are given in the supplementary materials, Table S1.
Section 2.3.:
- “The contents of phenolic acids and selected flavonoids were determined by UHPLC on Zorbax RRHD Eclipse…” Selected flavonoids? What does this mean? Does it mean that you focused on only some flavonoids? If so, this is not fine. You may have not identified a myriad of other compounds of this kind.
A: in the analysis we focused only on the three flavonoids mentioned. Based on the recommendation and due to lack of time for repeated measurements due to the addition of other flavonoids, we removed the information from the manuscript.
- RESULTS:
- All you presented here is not actually a novelty. It is obvious that the best results are given by embryonic organs in Spring since this is the right time when to collect them. You did not add anything new. In fact, my question is: why make this autumn collection? I think your work should have been focused mainly on the environmental and locality influence than on the collection time of the plant material. You would have done better to make a study more related to the interaction between buds and the environment.
A: On the recommendation we focused more on the influence of the locality on changes in phenolic metabolism. Changes in the time collected are mentioned only marginally. As a result, the title was changed: "The influence of locality on phenolic profile and antioxidant capacity of bud extracts”.
- And what about the altitudes of the collection sites? Was it important?
- And what about the correlation between altitude and the biosynthesis of secondary metabolites? I think you are aware that also altitude affects this biosynthesis. Was it important, too?
A: Analysis of the relationship between metabolite levels and altitude was included in the manuscript. Due to the relatively low range of altitude within our studied localities (250-330 m), we do not consider this variable important.
- Also, the decrease of some secondary metabolites from autumn to spring can be simply explained. By the way, you have embryonic organs which in Spring did not have enough time to fully biosynthesize flavonoids, tannins, and other classes. In this sense, why did you study these exact classes of natural compounds? Actually, I would not choose to collect embryonic organs if I were interested in isolating and identifying flavonoids, tannins and so on.
A: In view of the new findings, we have excluded the data from the manuscript, although in many publications these data have been taken into account.
- How do you explain all the antioxidant results you reported? What is the relation between the collection site and the antioxidant activities? In addition, according to the phytochemical pattern you reported, I would have expected opposite results for what concerns the antioxidant activities. How do you explain them?
A: The antioxidant system is very complex and involves various mechanisms, both enzymatic and non-enzymatic. In addition to the phenols studied, non-enzymatic antioxidants include other substances such as ascorbate, glutathione, tocopherols, carotenoids, proline, or sugars. Changes in these metabolites have not been analyzed, but their degree of involvement in antioxidant mechanisms may be high, which may lead to skewed results in individual antioxidant tests.
- CONCLUSIONS:
- This section should be revised. You did not fully demonstrate your conclusions.
- The section has been revised.

Round 2
Reviewer 2 Report
The authors improved the manuscript according to suggestions
Author Response
We would like to thank the Reviewer for valuable comments that helped us to improve the manuscript.

Reviewer 3 Report
The authors presented a revised version of the manuscript I have previously reviewed.
The manuscript has improved since my last view but not enough, in my opinion, to be accepted in its present form.
The things to revise and/or check again are listed below one by one:
- “Gemmotherapy represents a new type of phytotherapy using extracts made from embryonic parts of plants, i.e. buds, young sprout, leaves, or roots.” This sentence is not correct. No leaves and it is young roots and no roots in general. Please modify this part accordingly.
- Actually, it’s just not a good idea to collect plants rich in pollutants. Why did you?
- “The commercially available extracts of the monitored trees alone showed significantly lower values compared to our measurements (with the exception of oak for flavonoids).” And these values are…?
- How do you possibly explain the exception of flavonoids in oak?
- What about the comparison of your antioxidant results with others concerning your same studied species collected in other areas of the world? It would be interesting to see this.
- Some references are still incomplete. Missing pages.
Author Response
We have revised the manuscript: “The influence of locality on phenolic profile and antioxidant capacity of bud extracts” according to the comments and recommendations of the reviewers. In the following we give our response to reviewers. The changes are made in the new version of the manuscript. We apologize to you so much for our missing sending of our revised manuscript because of state holidays. We will be very thankful for consideration of our revised manuscript.
On behalf of all authors,
Kovalikova
Reply to Reviewer 3.
We would like to thank the Reviewer for valuable comments that helped us to improve the manuscript.
In following we list the changes made in the paper according to recommended revisions.
- “Gemmotherapy represents a new type of phytotherapy using extracts made from embryonic parts of plants, i.e. buds, young sprout, leaves, or roots.” This sentence is not correct. No leaves and it is young roots and no roots in general. Please modify this part accordingly.
A: This part was modified as follow: “Gemmotherapy represents the most recent therapeutic technique using properties of extracts made by the maceration in ethanol and glycerol of fresh meristematic plant tissues, mainly buds and sprouts.”
- Actually, it’s just not a good idea to collect plants rich in pollutants. Why did you?
A: Our aim is explained on page 2, lines 49-53; “Due to the good availability of macerating agents and the easiness of extract preparation, the production of gemmotherapeutic preparations is becoming accessible to the general public. The quality of these preparations may be influenced by the genotype and variety of the plant species, the phenological stage of the buds and, last but not least, the environmental characteristics of the sampling locality.”
- “The commercially available extracts of the monitored trees alone showed significantly lower values compared to our measurements (with the exception of oak for flavonoids).” And these values are…?
A: The passage in question was rewritten in the manuscript; see lines 283-291.
- How do you possibly explain the exception of flavonoids in oak?
A: Changes in case of flavonoids in oak can be explained by the effects of several factors. The genotype and phenotype of the selected trees itself may play a crucial role. Levels may also fluctuate due to weather conditions both in the previous growing season and in the harvest period. Furthermore, we can mention the different influence of environmental pollutants on the biosynthetic pathway of flavonoids, various enzymes involved in the synthesis may show different sensitivity to abiotic and biotic conditions of the environment, which is reflected in changes in their accumulation.
- What about the comparison of your antioxidant results with others concerning your same studied species collected in other areas of the world? It would be interesting to see this.
A: Additional information has been added to the text.
- Some references are still incomplete. Missing pages.
A: The bibliography was checked in detail and missing data were added.
